# Using publicly available satellite imagery and deep learning to understand economic well-being in Africa

Christopher Yeh [1,7], Anthony Perez [1,2,7], Anne Driscoll[3], George Azzari[2,4], Zhongyi Tang[5], David Lobell[3,4,5], Stefano Ermon [1] & Marshall Burke [3,4,5,6✉]

Accurate and comprehensive measurements of economic well-being are fundamental inputs into both research and policy, but such measures are unavailable at a local level in many parts of the world. Here we train deep learning models to predict survey-based estimates of asset wealth across ~ 20,000 African villages from publicly-available multispectral satellite imagery. Models can explain 70% of the variation in ground-measured village wealth in countries where the model was not trained, outperforming previous benchmarks from high-resolution imagery, and comparison with independent wealth measurements from censuses suggests that errors in satellite estimates are comparable to errors in existing ground data. Satellite-based estimates can also explain up to 50% of the variation in district-aggregated changes in wealth over time, with daytime imagery particularly useful in this task. We demonstrate the utility of satellite-based estimates for research and policy, and demonstrate their scalability by creating a wealth map for Africa's most populous country.

[1] Department of Computer Science, Stanford University, 353 Serra Mall, Stanford, CA 94305, USA. [2] AtlasAI, 459 Hamilton Ave, Palo Alto, CA 94301, USA. [3] Center on Food Security and the Environment, Stanford University, 616 Jane Stanford Way, Stanford, CA 94305, USA. [4] Department of Earth System Science, Stanford University, 473 Via Ortega, Stanford, CA 94305, USA. [5] Stanford Institute for Economic Policy Research, Stanford University, 366 Galvez St, Stanford, CA 94305, USA. [6] National Bureau of Economic Research, 1050 Massachusetts Avenue, Cambridge, MA 02138-5398, USA. [7]These authors contributed equally: Christopher Yeh, Anthony Perez. ✉email: mburke@stanford.edu

Local-level measurements of human well-being are important for informing public service delivery and policy choices by governments, for targeting and evaluating livelihood programs by governmental and non-governmental organizations, and for the development and deployment of new products and services by the private sector. While recent work has generated granular estimates of a range of human and physical capital measures in parts of the developing world[1–5], similar data on key economic indicators remain lacking, constraining even basic efforts to characterize who and where the poor are.

For example, at least 4 years pass between nationally representative consumption or asset wealth surveys in the majority of African countries (Fig. 1a), the key source of data for internationally comparable poverty measurements. These surveys have limited repeated observation of individual locations, making it difficult to measure local changes in well-being over time, and public release of any disaggregated consumption data from African countries is very rare. At current survey frequencies, we calculate that a given African household will appear in a household well-being survey less than once every 1000 years, or about two orders of magnitude less frequently than a household in the United States (Fig. 1b). While not all households need to be observed to generate accurate economic estimates, sampling enough households to generate frequent and reliable national-level statistics is alone likely to be expensive, requiring an estimated $1 billion USD annual investment in lower-income countries to measure a range of indicators relevant to the Sustainable Development Goals[6]. Expanding these efforts to generate reliable estimates at the local level would add dramatically to these costs.

Although existing data are scarce and traditional collection methods expensive to scale, other potentially relevant data for the measurement of well-being are being collected increasingly frequently. For instance, while most African households are never observed in consumption or wealth surveys, their location appears on average at least weekly in cloud-free imagery from multiple satellite-based sensors (Fig. 1b), and will have been observed in multispectral imagery at least annually for more than a decade.

Here we study whether such imagery can be used to accurately measure local-level well-being over both space and time in Africa. Earlier work demonstrated that coarse (1 km/pixel) nighttime lights imagery can measure country-level economic performance over time[7], and that high-resolution (<1 m/pixel) imagery from private-sector providers can be used to measure spatial variation in local economic outcomes in a handful of developing and middle-income countries[8–12]. Our focus is on using multiple sources of spatially coarser public imagery to infer both spatial and temporal differences in local-level economic well-being across sub-Saharan Africa, including for countries where reliable survey data do not yet exist and where survey-based interpolation methods might struggle to generate accurate estimates.

We find that a deep learning model trained on this imagery is able to explain ~70% of the spatial variation in ground-measured village-level asset wealth across Africa, and up to 50% of temporal variation when aggregating to the district level. We show that model performance is limited in large part by noise in the training data. We then demonstrate how our estimates could potentially be used to help target social programs and further understand the determinants of well-being across the developing world.

To develop our models, we assemble data on asset wealth for >500k households living in 19,669 villages across 23 countries in Africa, drawn from nationally representative Demographic and Health Surveys (DHS) conducted between the years 2009 and 2016 (Supplementary Fig. 1, Supplementary Table S1). We focus on asset wealth rather than other welfare measurements (e.g. consumption expenditure) as asset wealth is thought to be a less-noisy measure of households' longer-run economic well-being[13,14], is a common component of multi-dimensional poverty measures used by development practitioners around the world, is actively used as a means to target social programs[14,15], and is much more widely observed in publicly available georeferenced African survey data. Following standard approaches[13,16], for each household we compute a wealth index from the first principal component of survey responses to questions about ownership of specific assets ("Methods"). We pool all households in our sample in the principal components estimation such that the derived index is consistent over both space and time, and then average household values to the enumeration area level (also called clusters, and roughly equivalent to villages in rural areas or neighborhoods in urban areas), the level at which geocoordinates are available in the public survey data. This approach assumes that assets contribute similarly to wealth across all countries in our data. Alternative methods of constructing the index using

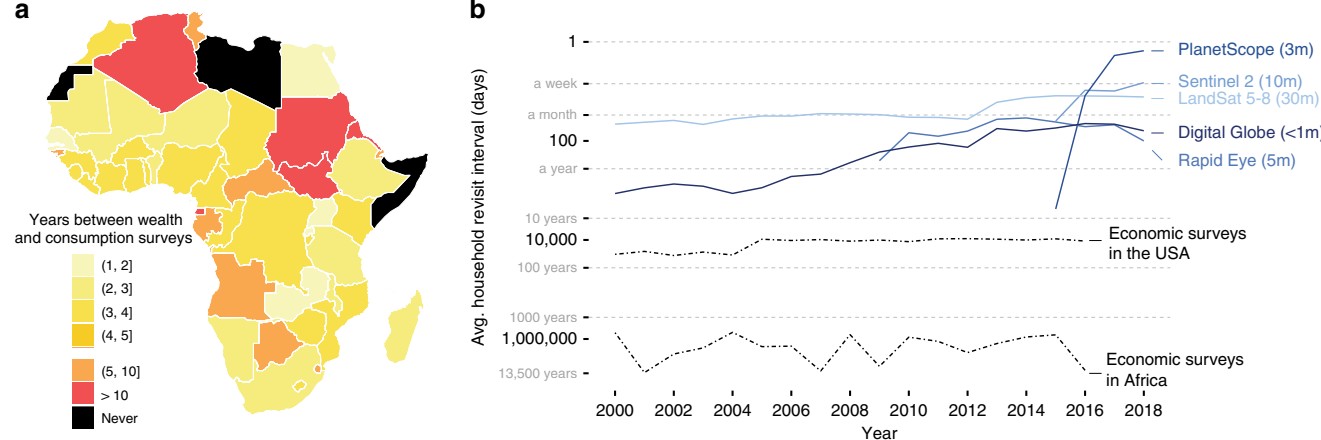

**Fig. 1 Economic data from household surveys are infrequent in many African countries. a** Frequency of nationally representative household consumption expenditure or asset wealth surveys across Africa, 2000–2016. **b** Average household revisit rate for surveys and average location revisit rate for various resolutions of satellite imagery over time. Survey revisit rate, the average time elapsed between observations of a given household in nationally representative expenditure or wealth surveys, is calculated as number of total person-days (population × 365) divided by the number of person-days observed in a given year. Satellite revisit rate estimates are calculated as the number of days in a year divided by the average number of images taken in a year across 500 randomly sampled DHS clusters in African countries, only counting images with <30% cloud cover.

only directly observable subsets of these assets, or which allow the mapping of assets to the wealth index to differ by country and year, yield very similar wealth estimates (Supplementary Fig. S2), and the wealth index is highly correlated with log consumption expenditure (weighted $r^2 = 0.5$, Supplementary Fig. S3) in a small subset of countries where consumption data are available.

We then train a convolutional neural network (CNN) to predict the village- and year-specific measure of wealth, using temporally and spatially matched multispectral daytime imagery from 30m/pixel Landsat and <1 km/pixel nighttime lights imagery as inputs (see "Methods"). Unlike earlier approaches that used nighttime light intensity as intermediate labels for training a CNN feature extractor on daytime imagery[8], we instead incorporate both sets of imagery in a deep learning model trained end-to-end, with models trained separately on daytime and nighttime images and then joined in a final fully connected layer. The goal of the model is to learn features in the daytime and nighttime imagery that are predictive of asset wealth, without first prescribing what features the model should look for. We compare performance of this dual-input combined model to models trained only on nightlights or on Landsat multispectral imagery, as well as to a transfer learning approach that uses nightlights as intermediate labels to find wealth-relevant features in the multispectral Landsat imagery[8]. We evaluate models using both pooled cross-country wealth data as well as only within-country data, and evaluate their ability to predict variation in well-being over space, and to predict changes over time. All evaluation is done on held-out test locations that the model did not use in training, an approach that limits overfitting as well as replicates the real-world setting of making predictions where ground data do not exist.

## Results

**Predictive performance over space.** Our combined model is predictive of cluster-level asset wealth, with predictions explaining on average 70% of the variation in ground-based wealth measurements in held-out country-years (Fig. 2a). Performance in individual held-out countries is never below 50% of variation explained, and often exceeds 80% (median = 70.4%, Fig. 2c, Supplementary Fig. S4), indicating our model is not simply separating wealthier African countries from poorer countries, but capably differentiating wealth levels within countries. These results exceed performance in earlier work on a similar task using high-resolution imagery[8,11] or mobile phone data[17] as input, and match or exceed benchmarks for in-country performance from geostatistical models used to predict health outcomes, standard of living, and housing quality in Africa[1,2,5,18] (see "Methods" for additional comparisons). Visualization of model-derived features suggests that the model learns semantically-meaningful features that are intuitively related to wealth, including filters for urban areas, agricultural regions, water bodies, and deserts (Supplementary Fig. S5). Aggregating predictions and ground measurements to the district level further improves performance (Fig. 2b, d), with predictions explaining on average 83% of the ground measurements in held-out countries not used to train the model. Improved performance with aggregation is consistent with errors cancelling when either the predictions or ground data are averaged.

Notably, CNNs trained only on nighttime lights (NL) or only on multispectral (MS) daytime imagery perform similarly to each other and almost as well as the combined model (MS+NL), suggesting that these two inputs contain similar information, at least for the task of predicting spatial variation in African wealth (Fig. 3a, b; Supplementary Fig. S6). Consequently, our approach of directly using nightlight images as model inputs performs better than using them indirectly as a proxy, as in an earlier

transfer learning approach[8]. Perhaps surprisingly, this trend holds even for highly data-limited settings: even when trained on data from only 5% of the surveyed clusters ($n < 1000$), our best models trained end-to-end outperform transfer learning (Fig. 3c).

A nearest neighbor model that predicts wealth in a given location from wealth in locations with similar nightlights values performs nearly as well as the deep learning models in predicting spatial variation, and much better than a linear model using scalar nightlights as input (Fig. 3a, b)—although neither nightlights models are predictive of temporal changes in wealth, while daytime models are (see below). These results suggest that non-linearities and/or spatial structure in nightlights is important for explaining spatial variation in well-being, and also may help explain why the transfer learning approach, which only predicts a scalar nighttime light intensity from daytime images, performs worse than end-to-end training.

**Predictive performance over time.** Many research and policy applications require estimates of changes in economic measures over time as well as over space. There are important challenges, however, in using available ground surveys to measure changes in economic outcomes over time at a local level, as well as in evaluating our deep learning approach's ability to do so. First, most existing surveys do not repeatedly measure outcomes at the same locations over time, i.e. they are not panel data; the DHS surveys, for instance, draw a new sample of clusters each survey round. Second, temporal changes over a few-year time span are likely to be small relative to cross-sectional differences, and any random noise in each year's survey will diminish the signal in these changes.

Given these challenges, we take three approaches to measuring and predicting changes in wealth over time. We first use repeated rounds of DHS surveys and spatially match a cluster in one survey year to the nearest cluster in a previous survey year (subject to the random noise added to the village locations by the survey team; see below), and compute wealth changes as the difference in wealth index between matched pairs of clusters. Second, we use an independent smaller set of household level panel data, the Living Standards Measurement Surveys (LSMS), to construct cluster-level changes in an asset wealth index. In both cases, predictions from a deep learning model using imagery as input can explain between 15% and 17% of the variation in survey-measured changes in asset wealth in held-out villages (Supplementary Fig. 7a, b). In contrast to our cross-sectional results, deep learning models using nightlights as input performed significantly worse than models using multispectral daytime imagery ($r^2 = 0.15$ vs. $r^2 < 0.01$), likely because nightlights show little variation over time in our sample locations (Supplementary Fig. S8). While temporal performance in multispectral models remains low relative to our model's performance in cross section, we show in a simulation that exceeding this temporal performance would be difficult for any model (Supplementary Fig. S9, Supplementary Note 1), as the small average temporal change in wealth in our sample (0.08 standard deviations of our wealth index) could easily be obscured by noise in the two survey values being differenced.

In a third experiment, we use the same LSMS data to construct a PCA-based index of changes in asset ownership (rather than a change in indexes, as before) to better capture the component of wealth that is actually changing. By construction this index has greater variation over time, and satellite-based features are more predictive ($r^2 = 0.35$), again with the models requiring multispectral daytime imagery inputs in order to perform well (Fig. 4a). As in the cross-sectional results, models again appear to learn features related to urbanization and to changing agricultural

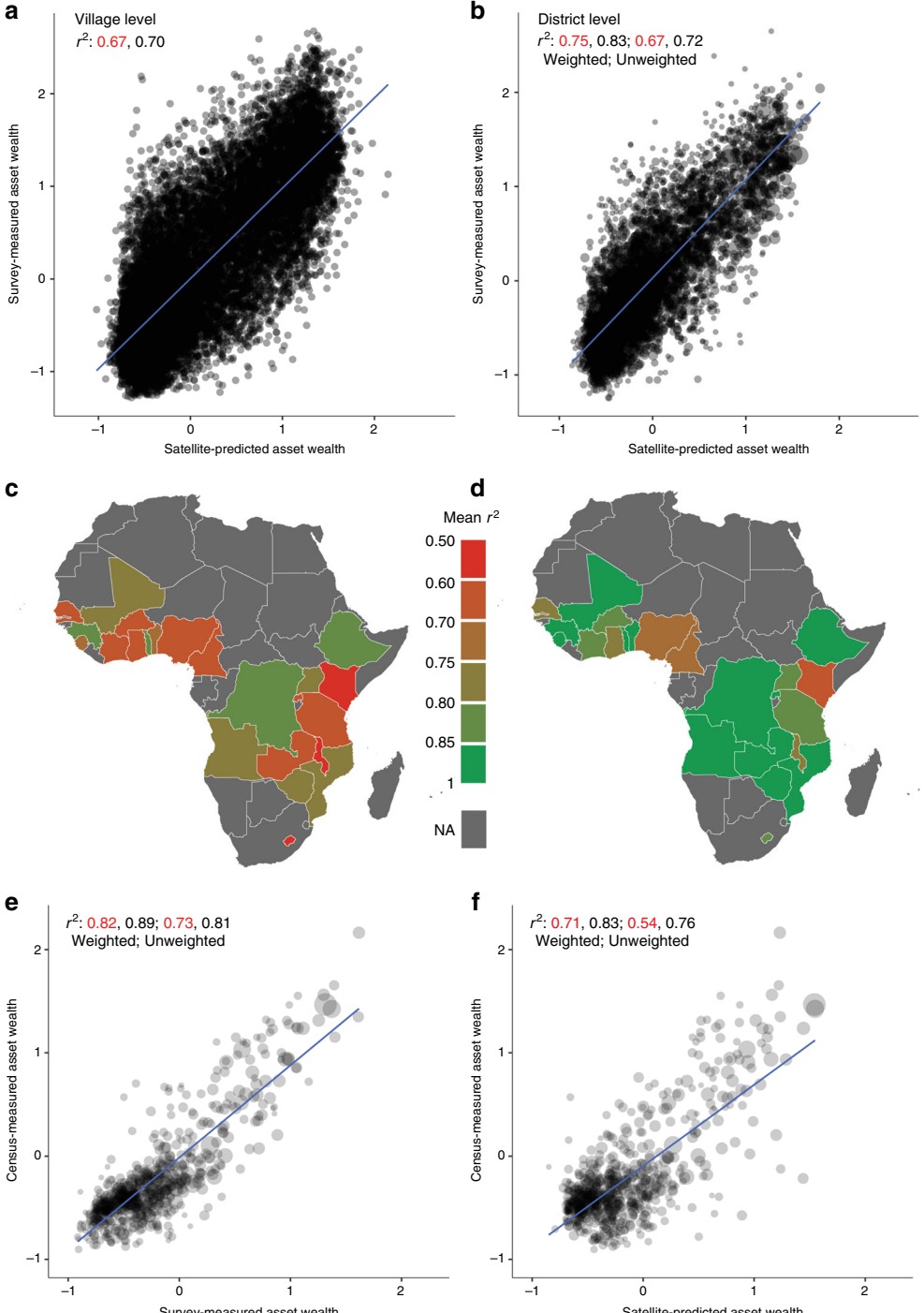

**Fig. 2 Satellite-based predictions explain the majority of variation in survey-based wealth estimates in all countries, and validate well against independent ground measures. a** Predicted wealth index versus DHS survey-measured wealth index across all locations and survey years; each point is a survey enumeration area (roughly, village) in a given survey-year, with satellite predictions generated by the CNN MS+NL model for each country from a model trained outside that country (predictions from 5-fold cross validation). $r^2$ values in red report goodness-of-fit on pooled observations, whereas values in black are the average of $r^2$ calculated within country-years. **b** As for **a**, but indices aggregated to the district level. **c** Average $r^2$ over survey years at the village level, by country. **d** Average $r^2$ over survey years at the district level, by country. **e** Comparison of DHS-based to independent census-based asset wealth at the district level, for available census measures within 4 years of DHS survey. **f** Comparison of asset wealth predicted by the CNN MS+NL model in held-out-countries to the independent census-based asset measures at the district level. In **b**, **e**, and **f**, $r^2$ is reported both weighted and unweighted by the number of villages contributing to each district-level average. The number of villages is represented by the dot size.

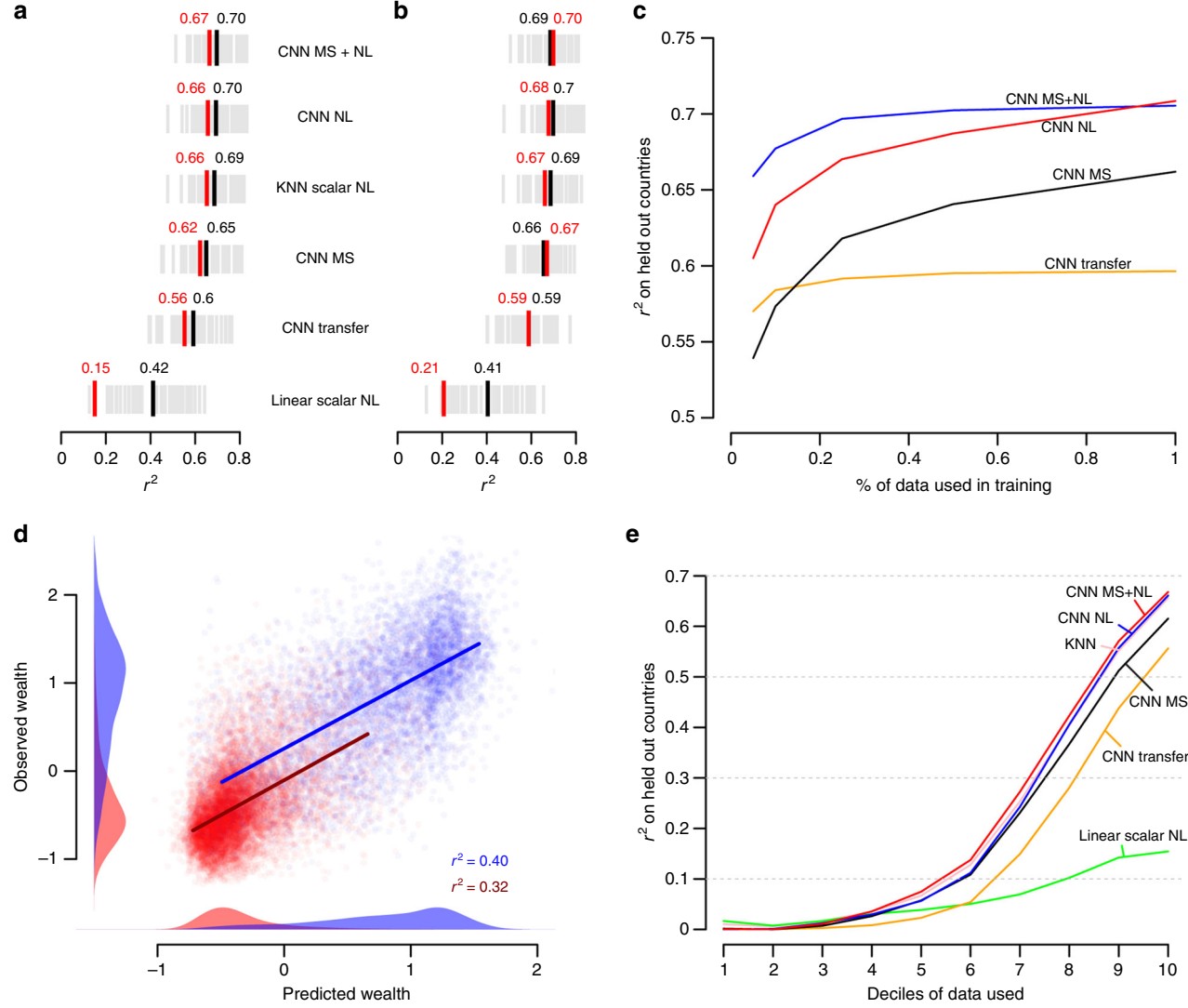

**Fig. 3 Performance by model and across different samples. a** Predictive performance of satellite predictions trained using 5 different machine learning models; `NL` nightlights, `MS` Landsat multispectral, and `transfer` transfer learning on nightlights with RGB Landsat imagery. Each grey line indicates the performance ($r^2$) on a held-out country-year, black lines and text show the average across country-years, and red lines and text show the $r^2$ on the pooled sample. **b** As in **a** but for evaluation on held-out villages within the same country. **c** Performance by amount of training data used. **d** Performance of `CNN MS+NL` model in urban versus rural regions in held-out countries. Model is trained on all data in training set and then applied separately to either urban or rural clusters in held-out countries. Each dot is an urban (blue) or rural (red) cluster, with densities showing the distribution of predicted (x-axis) and ground-measured (y-axis) wealth index. **e** Performance across the wealth distribution. Experiments were run separately for increasing percentages of the available clusters (e.g., x-axis value of 4 indicates that all clusters below 40th percentile in wealth were included in the test set).

patterns (Supplementary Fig. S10). Aggregating ground- and satellite-based estimates to the district level again leads to substantial performance improvements (Fig. 4b), with predictions of asset wealth changes explaining up to 50% of the ground-estimated changes in asset wealth. Improved performance with aggregation is again consistent with errors cancelling when either the predictions or ground data are averaged. To our knowledge, these are the first known remote-sensing based estimates of local-level changes in economic outcomes over time across a broad developing country geography, and provide benchmarks for future work.

**Understanding model performance.** While some of the combined model's overall performance in spatial prediction derives from distinguishing wealthier urban areas from poorer rural areas, the model is still able to distinguish variation in wealth within either rural or urban areas (Fig. 3d). In either case, much

of the model's explanatory power, at least in cross section, appears to be in separating wealthier clusters from poorer clusters rather than in separating the poor from the near poor (Fig. 3d). Performance at the country level (as shown in Fig. 2c) is not strongly related to country-level statistics on headcount poverty rates, urbanization, agriculture, or income inequality (Supplementary Fig. S11), although we do find that model performance is somewhat worse in settings where within-village variation in wealth is high. Poorer performance in these settings could be because our model has difficulty making accurate predictions in locally heterogeneous environments (a problem likely amplified by the random noise that has been added to the data; see below), or because sample-based estimates from the ground surveys are themselves more likely to be noisy when local variation is high.

Other sources of noise in the ground data (e.g. due to survey recall bias, sampling variation or geographic inaccuracies) could also worsen model performance. To explore the overall role of

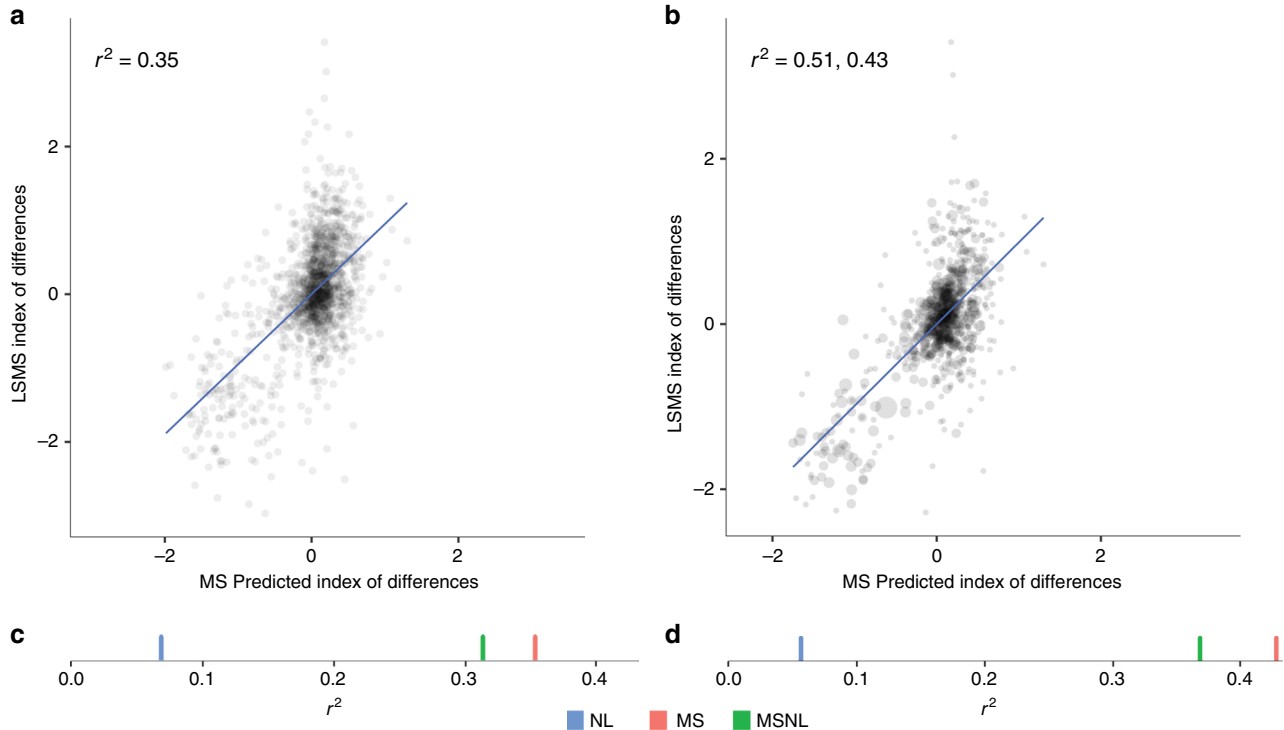

**Fig. 4 Satellite predictions of ground-measured changes in wealth over time. a** Performance of satellite-based model trained to predict the index of the change in wealth over time at the village level. The index is computed by finding the changes in assets at a household level and creating an index of those changes. Plot shows performance of model trained to predict this index of changes at the village level. **b** Same as a, but with observations aggregated to the district level. Dot size represents number of village observations in each district, and $r^2$ is reported both weighted ($r^2 = 0.51$) and unweighted by number of villages. **c, d** Cross-validated $r^2$ of models trained on multispectral (MS, red), nightlights (NL, blue), and both (MSNL, green). Every reported $r^2$ in **c, d** is unweighted.

ground-based error in model performance, we take two approaches. First, we compare both model-based and ground-based measures against an independent measure of asset wealth derived from census data in eight countries, with the comparison made at the district level, the lowest level of geographic identification available in public census data. We find that ground-based measures are only slightly more correlated with this independent wealth measure than our model-based estimates, and both are highly correlated with the independent estimate (Fig. 2e, f). This suggests that at least some of the prediction error in our main results derives from noise in the survey data.

Second, a known source of error in our ground data is the random noise added to village-level geo-coordinates by the survey implementers to protect privacy. In practice this jitter creates geographic misalignment between our input imagery and the true location of the surveyed villages; our approach is to look at all pixels in the $6.72 \times 6.72$ km neighborhood of the provided GPS location assuming that the village's true location falls in this neighborhood ($6.72$ km is the neighborhood defined by the input size of our CNN architecture and the pixel size of our imagery; see "Methods"), but much of this information might not be relevant to the specific village's asset wealth. To understand the performance cost of this noise, we iteratively add additional locational noise to our training data and then re-evaluate model performance on test data which are either also additionally jittered or not. Performance degrades with additional jitter (Supplementary Fig. S12), although much less rapidly when evaluating on data that have not also been additionally jittered. This suggests that the true (unobserved) performance of our main results is higher than we report, given that we are evaluating on data that have been jittered. Using these results to extrapolate backward to a hypothetical setting of no jitter in training data

suggests that locational noise in ground data is reducing model performance by $r^2 = 0.07$, or roughly the difference between our best and worst performing CNN models (Fig. 3).

**Downstream tasks.** To demonstrate the applicability of our satellite-based estimates to downstream research or policy tasks, we consider two use cases. The first is understanding why some locations are wealthier than others. Here we study associations between wealth and exposure to extreme temperatures, as much past work has indicated the wealth-temperature relationship is nonlinear[19,20], and because temperature data are readily available for all study locations in an independent gridded dataset[21]. Ground-based survey data indicate a non-linear relationship between village-level wealth and maximum temperature in the warmest month, and out-of-country estimates from CNN-based models recover this relationship very closely (Fig. 5a, "Methods"); estimates from simple scalar nightlights models do not. While none of these cross-sectional estimates are well suited for causal identification of the impact of temperature on wealth[19,22], we view the close match between satellite- and ground-based estimates of the temperature-wealth relationship as evidence that satellite-based estimates can be useful for these types of research questions.

We also use our estimates to evaluate the hypothetical targeting of a social protection program (e.g. a cash transfer), in which all villages below some asset level receive the program and villages above the threshold do not. Targeting on survey-derived asset data is a common approach to program disbursement in developing countries[23]. We compare targeting accuracy, defined as the percent of villages receiving the correct program, using estimates from different satellite-based models, under the assumption that survey-based ground data describe the true asset

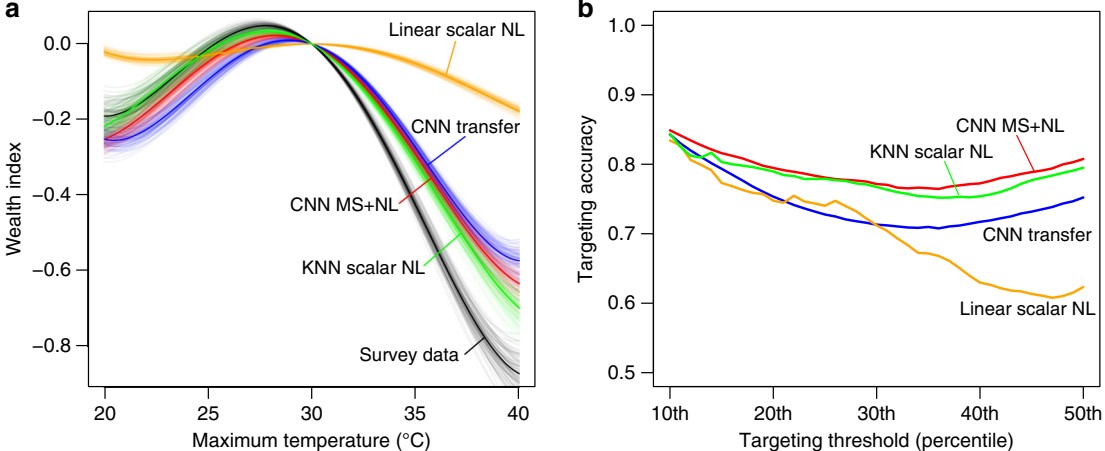

**Fig. 5 Using satellite-based wealth predictions in downstream tasks. a** Cross-sectional relationship between average maximum temperature and wealth across survey locations, as estimated with survey wealth data (black) and estimates from three satellite-based models. Each line is a bootstrap of the cross-sectional regression (100 bootstraps, sampling villages with replacement). Best-performing models recover temperature-wealth relationships that are closest to estimates using ground-measured data, and CNN-based models perform much better than scalar nightlights models. **b** Evaluation of a hypothetical targeting program in which all villages below some desired threshold in the asset distribution receive the program (e.g. a cash transfer) and villages above the threshold do not. We compare targeting accuracy, defined as the percent of villages receiving the correct program, using estimates from the same four satellite-based models as in **a**, under the assumption that survey-based ground data provide the true asset distribution. For instance, using MS+NL estimates to allocate a program to households below median wealth yields a targeting accuracy of 81%, versus 75% for CNN Transfer and 62% for scalar NL models. These estimates likely understate true targeting accuracy, given that ground data are themselves measured with some noise.

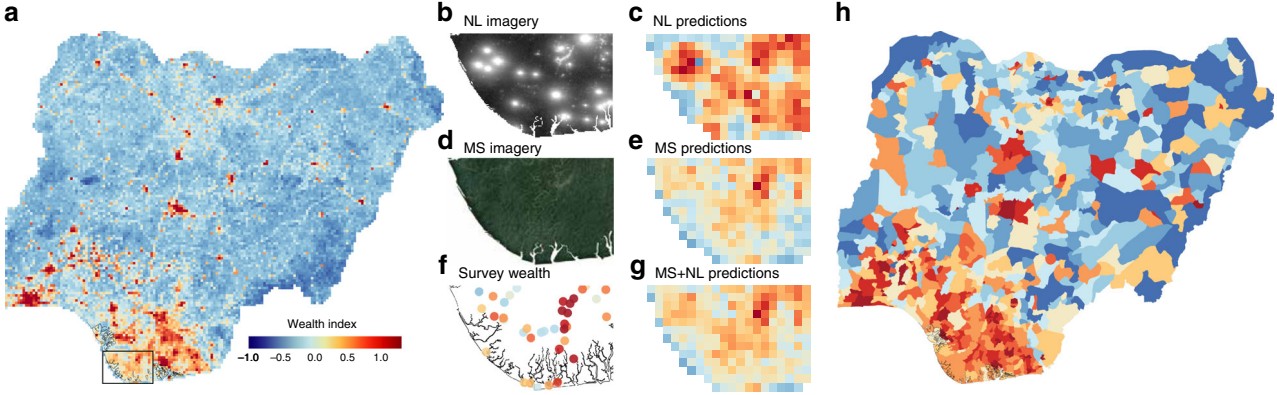

**Fig. 6 Spatial extent of imagery allows wealth predictions at scale. a** Satellite-based wealth estimates across Nigeria at pixel level. **b, d** Imagery inputs to model over region in Southern Nigeria depicted in box in **a**. **f** Ground truth input to model over the same region. **c, e, g** Model predictions with just nightlights (NL) as input, just multispectral (MS) imagery as input, and the concatenated NL and MS features as input. In this region, the model appears to rely more heavily on MS than NL inputs, ignoring light blooms from gas flares visible in **b**. **h** Deciles of satellite-based wealth index across Nigeria, population weighted using Global Human Settlement Layer population raster, and aggregated to Local Government Area level from the Database of Global Administrative Areas.

distribution. Our best performing satellite models again perform well on this task (Fig. 5b). For instance, using MS+NL estimates to allocate a program to households below median wealth yields a targeting accuracy of 81%, versus 75% for a CNN Transfer model and 62% for a scalar nightlights model. Importantly, these estimates likely understate true targeting accuracy given that ground data are themselves measured with some noise.

**Scalability**. To demonstrate the scalability of our overall approach, we construct a 7.65 km/pixel gridded wealth map of Nigeria, Africa's most populous country, for the years 2012–2014 using our model that combines daytime multispectral and nighttime imagery (Fig. 6). Visualizing both inputs and model predictions shows how our model learns to combine the two inputs, for example ignoring very bright nightlights pixels associated with oil flaring in the southern part of the country that are

not also associated with high wealth (Fig. 6b–g). Pixels are easily aggregated to higher administrative units using existing population rasters, and show strong latitudinal gradients of wealth across the country (Fig. 6h).

Generating the pixel-level raster involves processing ~9.1 billion pixels of daytime and nighttime imagery. Once the pipeline is developed, going from these raw imagery inputs to the prediction raster takes <30 h, including 4 h of model training on a NVIDIA Titan X GPU (excluding hyperparameter search), and roughly 24 h for imagery processing and raster generation. By comparison, a nationally representative household survey typically takes months to years to execute, at an average cost of $1–2 million USD[6]. While this comparison does not imply that our approach can replace household surveys, our approach can accelerate estimation of local-level wealth in years or in locations where survey data are unavailable.

## Discussion

Our satellite-based deep learning approach to measuring asset wealth is both accurate and scalable, and consistent performance on held-out countries suggests that it could be used to generate wealth estimates in countries where data are unavailable. Results suggest that such estimates could be used to help target social programs in data poor environments, as well as to understand the determinants of variation in well-being across the developing world.

However, while our CNN-based approach outperforms approaches to poverty prediction that use simpler features common in the literature (e.g. scalar nightlights[7]), the information the CNN is using to make a prediction is less interpretable than these simpler approaches, perhaps inhibiting adoption by the policy community. A key avenue for future research is in improving the interpretability of deep learning models in this context, and in developing approaches to navigate this apparent performance-interpretability tradeoff.

Our deep learning approach is also perhaps best viewed as a way to amplify rather than replace ground-based survey efforts, as local training data can often further improve model performance (Fig. 3b), and because other key livelihood outcomes often measured in surveys—such as how wealth is distributed within households, or between households within villages—are more difficult to observe in imagery. Similarly, our approach could also be applied to the measurement of other key outcomes, including consumption-based poverty metrics or other key livelihood indicators such as health outcomes. Performance in these related domains will depend both on the availability and quality of training data, which remains limited for key outcomes such as consumption in most geographies. Finally, our approach could likely be further improved by the incorporation of higher-resolution optical and radar imagery now becoming available at near daily frequency (Fig. 1b), or in combination with data from other passive sensors such as mobile phones[17] or social media platforms[24]. All represent scalable opportunities to expand the accuracy and timeliness of data on key economic indicators in the developing world, and could accelerate progress towards measuring and achieving global development goals.

## Methods

**Construction of asset wealth index.** The asset wealth index is constructed from responses to the set of questions about asset ownership that are common across DHS countries and waves: number of rooms occupied in a home, if the home has electricity, the quality of house floors, water supply and toilet, and ownership of a phone, radio, tv, car and motorbike. Variables such as floor type are converted from descriptions of the asset to a 1–5 score indicating the quality of the asset. We then construct an asset index at the household level from the first principal component of these survey responses, a standard approach in development economics[13,16]. This index is meant to capture household asset ownership as a single dimension, rather than act as a direct measure of poverty. By construction, the index has a mean equal to 0 and standard deviation of 1 across households. Supplementary Table S4 provides derived loadings for the first principal component.

Survey data are derived from 43 Demographic and Health Surveys (DHS) surveys conducted for 23 countries in Africa from 2009 to 2016 (Supplementary Table S1). In addition to the asset data, each DHS survey contains latitude/longitude coordinates for each survey enumeration area (or cluster) surveyed, each roughly equivalent to a village in rural areas and a neighborhood in urban areas. We removed clusters with invalid GPS coordinates and clusters for which we were unable to obtain satellite imagery, leaving us with 19,669 clusters. To protect the privacy of the surveyed households, DHS randomly displaces the GPS coordinates up to 2km for urban clusters and 10km for rural clusters[25]; this introduces a source of noise in our training data.

**Validating the wealth index.** The PCA-based index is quite robust to methods of calculation as well as variables included in the index. We compare our cross-country pooled PCA index to a measure that is the sum of all the assets owned, a PCA constructed from only objects that are owned (e.g. TV, radio) and not from housing quality scores which are more subjective, and country-specific asset indices created from running the PCA on each country separately. As shown in Supplementary Fig. S2, correlations between the pooled PCA index we use and these alternative variants range from $r^2 = 0.80$ to $r^2 = 0.98$.

**Replicating the wealth index in other contexts.** We then create similar asset indices using two separate external data sets: census data from countries whose censuses report asset ownership questions, and data from Living Standards Measurement Study (LSMS) conducted by the World Bank. In the publicly available census data, a 10 percent sample of microdata geolocated to the second administrative level (roughly, district or county) is available from each country. We focus on countries with public data who conducted censuses within 4 years of a DHS survey in our main sample and which had gathered data on assets similar to what was available in DHS. We found that 8 countries (Benin, Lesotho, Malawi, Rwanda, Sierra Leone, Senegal, Tanzania, and Zambia) had all asset variables used in DHS excluding motorbike and rooms per person. (Using DHS data, we find that the original index and an index constructed excluding these two variables had an $r^2 = 0.99$.) Our overall census sample yielded a total of 2,157,000 households observed in 656 administrative areas across these eight countries.

As census data are only georeferenced at second administrative levels, both DHS and census datasets are aggregated to the second-level administrative boundaries provided in the census data. Census data is aggregated using census household weights to construct representative district averages. A raw average across households is used to construct the corresponding DHS value; DHS and LSMS data do not provide household weights that allow construction of subnationally representative estimates.

We utilized asset wealth data from LSMS panel surveys for five countries (Malawi, Nigeria, Tanzania, Ethiopia, and Uganda). Cluster-level GPS coordinates are provided, with clusters in urban areas jittered up to 2 km and clusters in rural areas jittered up to 10 km. We are able to measure asset wealth for 9000 households over time in the LSMS data (roughly two orders of magnitude less than DHS), distributed over ~1400 clusters. As LSMS data follow households over time, we created a village-level panel using only households that existed in the first wave of interviews, removing any newly formed households or households that were not in later surveys. Additionally, where available, households that reported in the second survey that they had lived in their current location for less time than had elapsed since the first survey (i.e. migrant families) were removed. LSMS data were processed to try to match our DHS index as closely as possible, both by including the same assets and by matching asset quality definitions as similarly as possible. The fridge and motorbike variables were not available in the LSMS data and were excluded from the LSMS wealth index. Using DHS data, we find that the original index and an index constructed excluding the fridge and motorbike variables were highly correlated, with an $r^2$ of 0.974. While we cannot directly compare DHS and LSMS indices at the village level, district level estimates from the two sources have an $r^2$ of 0.60.

While our asset data cannot be used to directly construct poverty estimates—standard poverty measures are constructed from consumption expenditure data, which are not available in DHS surveys—household consumption aggregates are available in a subset of the LSMS data just described. Across six surveys in three countries, we find our constructed wealth index is fairly strongly correlated with log surveyed consumption at the village level, with a weighted $r^2$ of 0.50 (Supplementary Fig. S3). These results are consistent with findings that asset indices and consumption metrics are typically very comparable[14], and suggest that our approach to wealth prediction could perhaps be useful for consumption prediction as well, particularly as additional consumption data become available to train deep learning models.

**Satellite imagery.** We obtained Landsat surface reflectance and nighttime lights (nightlights) images centered on each cluster location, using the Landsat archives available on Google Earth Engine. We used 3-year median composite Landsat surface reflectance images of the African continent captured by the Landsat 5, Landsat 7, and Landsat 8 satellites. We chose three 3-year periods for compositing: 2009–11, 2012–14, and 2015–17. Each composite is created by taking the median of each cloud free pixel available during that period of 3 years. The motivation for using three-year composites was two-fold. First, multi-year median compositing has seen success in similar applications as a method to gather clear satellite imagery[26], and even in 1-year compositing we continued to note the substantial influence of clouds in some regions, given imperfections in the cloud mask. Second, the outcome we are trying to predict (wealth) tends to evolve slowly over time, and we similarly wanted our inputs not be distorted by seasonal or short-run variation. The images have a spatial resolution of 30 m/pixel with seven bands which we refer to as the multispectral (MS) bands: RED, GREEN, BLUE, NIR (Near Infrared), SWIR1 (Shortwave Infrared 1), SWIR2 (Shortwave Infrared 2), and TEMP1 (Thermal).

For comparability, we also created 3-year median composites for our nightlights imagery. Because no single satellite captured nightlights for all of 2009–2016, we used DMSP[27] for the 2009–11 composite, and VIIRS[28] for the 2012–14 and 2015–17 composites. DMSP nightlights have 30 arc-second/pixel resolution and are unitless, whereas VIIRS nightlights have 15 arc-second/pixel resolution and units of $nWcm^{-2}sr^{-1}$. The images are resized using nearest-neighbor upsampling to cover the same spatial area as the Landsat images. Because of the resolution difference and the incompatibility of their units, we treat the DMSP and VIIRS nightlights as separate image bands in our models.

Both MS and NL images were processed in and exported from Google Earth Engine[29] in 255 × 255 tiles, then center-cropped to 224 × 224, the input size of our CNN architecture, spanning 6.72 km on each side (30 m Landsat pixel size × 224

px $= 6.72$ km). Note that this means any survey cluster whose location coordinates are artificially displaced by more than 4.75 km ($6.72/\sqrt{2}$) is completely beyond the spatial extent of the satellite imagery. Each band is normalized to have mean 0 and standard deviation 1 across our entire dataset. The raster of wealth in Nigeria in Fig. 5 was generated by exporting non-overlapping tiles from Google Earth Engine, following the same processing steps as for model training.

**Deep learning models**. Our deep CNN models use the ResNet-18 architecture (v2, with preactivation)[30], chosen for its balance of compactness and high accuracy on the ImageNet image classification challenge[31]. We modify the first convolutional layer to accommodate multi-band satellite images, and we modify the final layer to output a scalar for regression. For predicting changes in wealth and the "index of differences" on the LSMS data, we stack together the images from two different years to create a $224 \times 224 \times (2C)$ image, where $C$ is the number of channels in a single satellite image.

The modifications to the first convolutional layer prevent direct initialization from weights pre-trained on ImageNet. Instead, we adopt the same-scaled initialization procedure[32]: weights for the RGB channels are initialized to values pre-trained on ImageNet, whereas weights for the non-RGB channels in the first convolutional layer are initialized to the mean of the weights from the RGB channels. Then all of these weights are scaled by $3/C$ where $C$ is the number of channels. The remaining layers of the ResNet are initialized to their ImageNet values, and the weights for the final layer are initialized randomly from a standard normal distribution truncated at $\pm 2$. For the models trained only on the nightlights bands, we initialized the first layer weights randomly using He initialization[33]. When predicting changes in wealth and when predicting the index of differences on the LSMS data, we used random initialization instead, as it performed better than using same-scaled ImageNet initialization on the validation sets (see "Cross-Validation").

The ResNet-18 models are trained with the Adam optimizer[34] and a mean squared-error loss function. The batch size is 64 and the learning rate is decayed by a factor of 0.96 after each epoch. The models are trained for 150 epochs (200 epochs for DHS out-of-country). The model with the highest $r^2$ on the validation set across all epochs is used as the final model for comparison. This is done as a regularization technique, equivalent to early-stopping. We performed a grid search over the learning rate (1e-2, 1e-3, 1e-4, 1e-5) and $L_2$ weight regularization (1e-0, 1e-1, 1e-2, 1e-3) hyperparameters to find the model that performs the best on the validation fold. To prevent overfitting, the images are augmented by random horizontal and vertical flips. The non-nightlights bands are also subject to random adjustments to brightness (up to 0.5 standard deviation change) and contrast (up to 25% change). Additionally, for predicting changes in wealth and the index of differences on the LSMS data, we randomize the order for stacking the satellite images (i.e. stacking the before image on top of or below the after image), multiplying the label by $-1$ whenever the after image was stacked on top to signify a reversed order.

When using the two nightlights bands, we set pixels in the non-present band to all zeros. This ensures that the first-layer weights for that band are not updated during back-propagation, because the gradient of the loss with respect to the weights for the all-zero band becomes zero. Furthermore, since the ResNet-18 architecture has a batch-normalization layer following each convolutional layer, there are no bias terms.

For models incorporating both Landsat and nightlights (i.e. our combined model), we trained two ResNet-18 models separately on the Landsat bands and nightlights bands, respectively, and joined the models in their final fully connected layer. In other words, we concatenated the final layers of the separate Landsat and Nightlights models and trained a ridge-regression model on top. We found that this approach performed better than stacking the nightlights and Landsat bands together in a single model.

For DHS data, an average of 25.59 households (standard deviation $= 5.59$) were surveyed for each village, compared to an average of 6.37 households (sd $= 3.57$) in LSMS. Due to the lower number of households surveyed for LSMS, which results in noisier estimates of village-level wealth, we weighted LSMS villages proportional to their surveyed household count in the loss function during training. We did not weight DHS villages.

**Transfer learning models**. We compared our end-to-end training procedure with the transfer learning approach first proposed by Jean et al.[8]. In this approach, nightlights are a noisy but globally available proxy for economic activity ($r^2 \approx 0.3$ with asset wealth), and a model is trained to predict nighttime lights values from daytime multispectral imagery. This process summarizes high-dimensional input daytime satellite images as lower-dimensional feature vectors than can then be used in a regularized regression to predict wealth.

Because our images have a mixture of DMSP and VIIRS values, and the two satellites have different spatial resolutions, the binning approach in Jean et al.[8] that treated nightlights prediction as a classification problem was unworkable. Instead, we framed transfer learning as a multitask regression problem. We extracted the neural network's final layer output predictions for both the DMSP value and the VIIRS value, and regressed on whichever nightlights label was available for each daytime image. On the nightlights prediction task over locations sampled from all 23 DMSP countries, our transfer learning models achieved performance of $r^2 = 0.82$ when using RGB bands and $r^2 = 0.90$ when using all Landsat bands; these values are not directly comparable to results in Jean et al.[8], as that work posed nightlights prediction as a

3-class classification problem. With these models trained to predict nightlights values from daytime imagery, we froze the model weights and fine-tuned the final fully connected layer to predict the wealth index. We note that our transfer learning experiments contain a much larger set of countries than the Jean et al.[8] results, which focused on five countries, and thus are not directly comparable.

**Baseline models**. We train simpler $k$-nearest neighbor models (KNN) on nightlights that predict wealth in a given location $i$ as the average wealth over the $k$ locations with nightlights values closest to that in $i$. In essence, this model allows a non-linear and non-monotonic mapping of nightlights to wealth. The hyperparameter $k$ is tuned by cross-validation. We also train a regularized linear regression on scalar nighlights (scalar NL) as a baseline model.

**Training on limited data**. To evaluate how models perform in even more data-limited situations, we trained our deep models on random subsets of 5%, 10%, 25%, 50%, and 100% of the full training data, repeated over 3 trials with different random subsets. For each subset size, we report the mean $r^2$ over the three trials (Fig. 3c).

**Data splits**. For both DHS and LSMS survey data, we split the data into 5 folds of roughly equal size for cross-validation. For the DHS out-of-country tests, we manually split the 23 countries into the 5 folds such that each fold had roughly the same number of villages, ranging from 3909 to 3963 (Supplementary Table S2). As described below, models were trained using cross-validation to select optimal hyperparameters. Each model was trained on 3-folds, validated on a 4th, and tested on a 5th. The fold splits used in the cross-validation procedure are shown in Supplementary Table S3. For DHS in-country training, we split the 19,699 villages into 5 folds such that there was no overlap in satellite images of the villages between any fold, where overlap is defined as any area (however small) that is present in both images. We used the DBSCAN algorithm to group together villages with overlapping satellite images, sorted the groups by the number of villages per group in decreasing order, then greedily assigned each group to the fold with the fewest villages. We followed the same procedure to create 5 LSMS in-country folds. We did not perform out-of-country tests with LSMS data.

**Cross-validation**. For each of the input band combinations (MS, MS+NL, NL), we trained five separate models, each with a different test fold. Of the four remaining folds, three folds were used to train the models, with the final fold designated as the validation set used for early stopping and tuning other hyperparameters (Supplementary Table S3). Once the CNNs were trained, we fine-tuned the last fully connected layer using ridge regression with leave-one-group-out cross-validation. In the out-of-country setting, we fine-tuned the final layer individually for each test country, using data from all other countries. Thus, the convolutional layers in the CNNs have effectively seen data from four of the 5-folds, while the final layer sees data from every country except the test country. In the in-country setting, we only used data from the non-test folds for fine-tuning.

Ideally, the hyperparameters for machine learning models should be tuned by cross-validation for optimal generalization performance on unseen data. However, because training deep neural networks requires substantial computational resources, leave-one-group-out cross-validation is prohibitively time intensive (where in our setting, each group is a country). Consequently, we performed leave-one-fold-out cross-validation for all the hyperparameters for the body of the CNN, and only used leave-one-group-out cross-validation to tune the regularization parameter for training the weights in the final fully connected layer.

**Comparison with previous benchmarks**. Our model achieves a cross-validated $r^2 = 0.67$ on pooled cluster-level observations in held-out countries (or $r^2 = 0.70$ when averaging over $r^2$ values from each country). This meets or exceeds published performance on related tasks, including using high-resolution imagery and transfer learning to predict asset wealth in five African countries[8] ($r^2 = 0.56$), using call detail records to predict asset wealth in Rwanda[17] ($r^2 = 0.62$), and using survey data and geospatial covariates to predict housing quality[5] ($r^2 = 0.67$), child stunting[1] ($r^2 = 0.49$), diarrheal incidence[2] ($r^2 = 0.47$ averaged over years) across sub-Saharan Africa or to predict standard of living in Senegal[18] ($r^2 = 0.69$). All values are for published cross-validated performance at the cluster or pixel level (except for diarrheal incidence whose performance is only reported at the admin-2 level).

As our primary focus is on constructing and evaluating out-of-country predictions, our results are not directly comparable to findings from other small area estimate approaches that rely on having in-country surveys with which to extract covariates and make local-level predictions (e.g. refs. [35,36]). However, our satellite-derived wealth estimates and/or the satellite-derived features themselves could be used as input to these small area estimates, and evaluating the utility of satellite-derived data in such settings is a promising avenue for future research.

**Research and policy experiments**. To study whether our satellite-based estimates can be used to shed light on the determinants of the spatial distribution of wealth—a longstanding research question—we match our ground-based and satellite-based wealth estimates to gridded data on maximum temperature in the warmest month[21]. We study temperature as our potential wealth determinant because past

work has suggested that differences in temperature exert significant, non-linear influence on economic output[19,20], because temperature data are readily available for all our study locations.

We extract the maximum average monthly temperature for each cluster in our dataset (averaged over the years 1970–2000[21]) and then flexibly regress wealth estimates on temperature:

$$w_i = f(T_i) + \varepsilon_i \qquad (1)$$

where $w_i$ is the wealth estimate for cluster $i$ and $f(T_i)$ is a fourth-order polynomial in temperature. To capture uncertainty in our estimates of $f()$, we bootstrap Eq. (1) 100 times for each different wealth measure, sampling villages with replacement. We compare estimates of $f()$ when we measure $w_i$ using the ground data or when using various satellite-based estimates: our benchmark MS + NL estimates, or the two main other published approaches, CNN transfer learning[8] and scalar nightlights[7]. Results are shown in Fig. 5a. We emphasize that these cross-sectional estimates of $f()$ do not represent causal estimates of the impact of temperature on wealth, as many other factors are known to be correlated with both temperature and wealth (e.g. institutional quality, disease environment, nearby trading partners, etc.)[22].

To study whether our satellite-based estimates can be used for policy tasks, we evaluate the hypothetical targeting of a social protection program (e.g. a cash transfer), in which all villages below some asset level receive the program and villages above that level do not. Such targeting on survey-derived asset data is a common approach to program disbursement in developing countries[23]. Because asset indices constitute a relative measure of wealth and it is not obvious how to set an absolute cut-off to define who is poor, standard practice is instead to divide the population into percentiles in the asset distribution and then designate bottom percentiles as poor[15].

We follow that practice here. Using the ground data, we define a threshold $w_{p,g}^*$ corresponding to a chosen percentile $p$ in the ground-measured asset distribution, and designate any village with wealth below that threshold as a program beneficiary (a treated village), i.e. $t_{i,g,p} = \mathbb{1}[w_{i,g} < w_{p,g}^*]$, where $w_{i,g}$ is village $i$'s measured wealth in the ground data and $t_{i,g,p}$ denotes that villages treatment status according to the ground data. We then follow the same procedure for a satellite-estimated wealth distribution $s$, choosing the same percentile $p$ in the satellite-estimated distribution to define treatment. This yields each village's treatment status under the satellite-derived estimates $t_{i,s,p} = \mathbb{1}[w_{i,s} < w_{p,s}^*]$. We note that we are fixing $p$ between ground- and satellite-based estimates rather than fixing the wealth threshold, such that the same overall number of villages are treated in both the ground-measured case and the satellite-measured case.

Under the assumption that the ground-derived treatment statuses $t_{i,g}$ are correct, we then define targeting accuracy $A_{s,p}$ as the proportion of satellite-derived treatment statuses that are correct under a given percentile cutoff $p$, i.e. $A_{s,p} = \frac{1}{n} \sum_{i=1}^{n} \mathbb{1}[t_{i,s,p} = t_{i,g,p}]$, where $n$ is the total number of villages. We compute $A_s$ under different values of $p$ ranging from the 10th to the 50th percentile, and for the same three different satellite-based wealth estimates $s$ (MS+NL, transfer learning, and scalar NL) used in Fig. 5. We emphasize that to the extent that the ground data $w_{i,g}$ are measured with noise, which we have strong evidence of (see Supplementary Fig. S12 and Fig. 2e, f), our estimated targeting accuracy likely understates true targeting accuracy.

**Reporting summary**. Further information on research design is available in the Nature Research Reporting Summary linked to this article.

## Data availability
Data to replicate all findings in the paper are available at https://github.com/sustainlab-group/africa_poverty.

## Code availability
Code to replicate all findings in the paper are available at https://github.com/sustainlab-group/africa_poverty.

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

## Acknowledgements

We thank USAID Bureau for Food Security and the Stanford King Center on Global Development for funding support. Ermon acknowledges National Science Foundation grants #1651565 and #1522054 for additional support.

## Author contributions

M.B., S.E., and D.L. conceived of the study. G.A. and Z.T. processed and analyzed satellite imagery. A.D. processed and analyzed survey and census data. C.Y. and A.P. trained deep learning models and analyzed their output with A.D. and M.B. C.Y., A.P., A.D., D.L., S.E., and M.B. interpreted results of model experiments. A.P. and C.Y. generated the wealth raster. C.Y., A.P., A.D., S.E., D.L., and M.B. wrote the paper.

## Competing interests

M.B., D.L., and S.E. are co-founders of AtlasAI, a company that uses machine learning to measure economic outcomes in the developing world. A.P. and G.A. conducted the research for this paper while students or employees at Stanford, but are also now employed at AtlasAI.
