## [Peer Review File · Nature Communications]

Reviewers' comments:

Reviewer #1 (Remarks to the Author):

Overall a really interesting paper. I have added comments within the attached pdf. Some of these refer to issues that I would recommend are dealt with as part of the edits. The purpose of the paper is interesting and worthwhile and it potentially has a broad appeal. However, currently the writing style is very focused on data science with technical language and jargon used too frequently. This will reduce the number of people that will be able to engage fully with this work which is a shame as it should be broadly appealing. I have below provided answers to each of the questions that Nature asks to be considered. I have also included additional comments within the pdf. Some of these may be difficult to understand as they are notes to myself during the review process. But I have left them in as they may help the authors understand how a reader interpreted some aspects of the work.

- What are the major claims of the paper?

The paper investigates the role that EO data can have in predicting well-being across 23 countries in Sub-Saharan Africa. Its main claim is that the prediction accuracy is 70% for village level well-being and increases to over 80% when data are aggregated to districts.

- Are the claims novel? If not, please identify the major papers that compromise novelty

The claims are similar to other papers that have been published in the past couple of years. The accuracy reported here is slightly higher than other past papers. The novelty is that this paper uses a very large number of households across different time periods and multiple different countries.

- Will the paper be of interest to others in the field?

This paper will be of interest to others in the field. I work in the field and I am interested in the results. However, in its current form it is quite technical and contains substantial levels of jargon making it difficult for people that are not data scientists to understand what was done.

- Will the paper influence thinking in the field?

I think it will influence thinking in terms of what is possible. However, it currently leaves more questions than answers. For example, the 70% accuracy is for a few select countries. The majority are lower than this (around 50-60% which is more similar to other papers in the field). There is not enough discussion within the paper about what may be driving the variation in accuracies between different countries.

- Are the claims convincing? If not, what further evidence is needed?

The claims are not that convincing at the moment considering the wide audience that Nature Communications papers will appeal to. The claims will be convincing to data scientists but not policy makers or others working in the development field. There is increasing interest from development practitioners about the role that EO can play in predicting poverty/well-being or wealth and how this can support current work, monitoring or SDGs etc. However this is not clear enough in the paper in its current form.

- Are there other experiments that would strengthen the paper further? How much would they improve it, and how difficult are they likely to be?

There was a lot of additional statistical analyses done to examine accuracy. However, I would prefer to see either some additional analysis or discussion around the following two points:

1. Why the asset index weights were derived from the entire sample and no consideration was given to deriving weights from individual countries or different time periods. The asset indices could look very different if they were constructed in different ways and therefore this could have significant

impacts on the predictive outcomes. If a policy maker was to look at this work currently it would appear that they could achieve 70% accuracy. However, if they wanted to repeat the analysis for a specific country the accuracy they would get would be different.

2. Its currently not clear to a wide enough audience how the change in assets over time are related to changes in EO data metrics. If this could be focused on it would be were novel as previous works by other researchers have not looked at the relationships between changes in poverty/well-being and EO data

- Are the claims appropriately discussed in the context of previous literature?

My major question about the claims is that Why only use Median reflectance over 3 years from Landsat? What theoretical basis is there for reflectance being linked to poverty. In other studies land use or land cover is used which is related to poverty by for example saying that building roof material is an indicator of well-being. Policy makers like this approach as it can be related to literature and understanding and other policies they are working on at the time. With reflectance and median reflectance there is less clarity on how it is linked. I know that data science is not as concerned by this but I think some discussion about this would certainly broaden the appeal of the paper.

The other previous literature is covered mostly as far as I am aware. But the change over time could be emphasised a little more.

- If the manuscript is unacceptable in its present form, does the study seem sufficiently promising that the authors should be encouraged to consider a resubmission in the future?

Yes, I would say that a resubmission is an excellent approach here as it is certainly worth considering once the clarity of the message is improved.

Reviewer #2 (Remarks to the Author):

Review of the manuscript entitled "Using publicly available satellite imagery and deep learning to measure and understand economic well-being in Africa." This manuscript seeks to use publicly available remote sensing data to map economic well being in Africa. The major claims of this paper are that it provides a state of the art methodology for mapping poverty over space and time in multiple African countries. Their results, while similar to their previous results in Jean et al. 2016, appear to be slightly better and use data sets that are more conducive to global research. Additionally, they have added a change in wealth over time component by performing a panel analysis and have expanded to more countries. These results are novel and are of interest to a wide array of researchers within the field. This paper would influence thinking within the field of big data and poverty as this is a first at this spatial scale and over time. There are some major and minor comments that do arise and are given below. The major issue with this paper is that they are using a vague representation of both wealth and the values they are extracting from remotely sensed images. These issues are described in questions 1, 2, and 3 below. This does not allow a policy maker to really understand what is driving poverty and actually poverty levels. Finally, I don't feel confident in evaluating the appropriateness and validity of the statistical analysis within the data and suggest that an expert in CNNs evaluate this manuscript.

Major questions:

1. How does their asset index compare to consumption estimates of poverty that the World Bank generally uses? Can they compare their estimates to at least one country so that it can be known what the differences are in these two methods of poverty estimations and how their results would compare?
2. Sustainable Development Goal #1.1 is to eradicate extreme poverty everywhere. This is currently measured as people living on less than \$1.25 a day. Can the results from this study be used to

determine where these people are and if they are being lifted out of extreme poverty?

3. What values extracted from the satellite data can be used to understand what the model is using to predict variability in poverty so that policy makers can make informed decisions?

4. Why is a 6.72 km pixel used in this analysis? Also, if this is the pixel size used, why does the map of Nigeria in figure 4, have 7.5 km pixels?

5. Why are three years of Landsat data used to cover an area? That appears to be long time in rapidly changing countries such as these.

6. Why were just DHS data compared out of country and not LSMS? Also, how comparable are the DHS and LSMS data sets? It appears that multiple countries have both. Can one be predicted with the other?

7. In the results, Kenya appears to have the poorest performance. Can this be explained?

8. When the Night-time lights and Landsat CNN models are used to create their "combined" model using ridge regression (p. 10 lines 331-336), how many independent variables are there going into the regression? Why is Ridge regression used versus Elastic Net that incorporates LASSO as well?

9. The only statistical values that are given for model performance are r-squared. What are the model error estimates and how do they vary spatially and between countries?

Minor comments:

1. In Figure s1, use a different range of colors so wealthy and poor can be more easily identified.

2. In Figure 4, where are the Local Government Area boundaries and population data from used to create this figure?

Reviewer #1 (Remarks to the Author):

Overall a really interesting paper. I have added comments within the attached pdf. Some of these refer to issues that I would recommend are dealt with as part of the edits. The purpose of the paper is interesting and worthwhile and it potentially has a broad appeal. However, currently the writing style is very focused on data science with technical language and jargon used too frequently. This will reduce the number of people that will be able to engage fully with this work which is a shame as it should be broadly appealing. I have below provided answers to each of the questions that Nature asks to be considered. I have also included addition comments within the pdf. Some of these may be difficult to understand as they are notes to myself during the review process. But I have left them in as they may help the authors understand how a reader interpreted some aspects of the work.

Response: We appreciate the reviewer's kind words and have reworked the text to eliminate all the jargon that we can, and have attempted to define all terms more clearly. We have also gone through R1's line edits/suggestions in the pdf that s/he provide and incorporated those as well. We hope this increases the overall readability and general appeal of the paper!

• What are the major claims of the paper?

The paper investigates the role that EO data can have in predicting well-being across 23 countries in Sub-Saharan Africa. Its main claim is that the prediction accuracy is 70% for village level well-being and increases to over 80% when data are aggregated to districts.

• Are the claims novel? If not, please identify the major papers that compromise novelty

The claims are similar to other papers that have been published in the past couple of years. The accuracy reported here is slightly higher than other past papers. The novelty is that this paper uses a very large number of households across different time periods and multiple different countries.

• Will the paper be of interest to others in the field?

This paper will be of interest to others in the field. I work in the field an I am interested in the results. However, in its current form it is quite technical and contains substantial levels of jargon making it difficult for people that are no data scientists to understand what was done.

Response: We appreciate this comment and have reworked the text in our best effort to eliminate all jargon and define terms as appropriate.

• Will the paper influence thinking in the field?

I think it will influence thinking in terms of what is possible. However, it currently leaves more questions than answers. For example, the 70% accuracy is for a few select countries. The majority are lower than this (around 50-60% which is more similar to other papers in the field). There is not enough discussion within the paper about what may be driving the variation in accuracies between different countries.

Response: We note that the median accuracy across countries is 70.4%, so fully half of our countries exceed this number. We have now clarified this in the main text.

Regarding the variation in accuracy across countries, we agree that this is an interesting question. In response, we now directly analyze what factors drive the variation in predictive performance of our model

across countries. In particular, we take the country-specific predictive performances shown in Fig 2c, and regress them against country-level measures of GDP per capita, headcount poverty rate, GINI index, % income from agriculture, total population, and % of population in urban areas. We also compare prediction error against country-level estimates of the magnitude of within-village wealth variance compared to overall wealth variance -- essentially a measure of the amount of overall variation in wealth in a country that is from within-village (as opposed to between village) differences. Results are shown in new Fig S11, and summarized in the main text as follows:

“Performance at the country level (as shown in Fig. 2c) is not strongly related to country-level statistics on poverty, urbanization, agriculture, or income inequality (Fig. S11), although we do find that model performance is somewhat worse in settings where within-village variation in wealth is high. Poorer performance in these settings could be because our model has difficulty making accurate predictions in locally heterogeneous environments, or because sample-based estimates from the ground surveys are themselves more likely to be noisy when local variation is high.”

• Are the claims convincing? If not, what further evidence is needed?

The claims are not that convincing at the moment considering the wide audience that Nature Communications papers will appeal to. The claims will be convincing to data scientists but not policy makers or others working in the development field. There is increasing interest from development practitioners about the role that EO can play in predicting poverty/well-being or wealth and how this can support current work, monitoring or SDGs etc. However this is not clear enough in the paper in its current form.

Response: The referee seems primarily concerned about the relevance of our paper to the broader policy audience. In response, we have reduced jargon to every extent possible as well as added additional text to the discussion highlighting lessons for both practitioners and policymakers who need to understand the usefulness of our approach and data in light of data they might already be collecting. The concluding discussion section now reads:

“Our satellite-based deep learning approach to measuring asset wealth is both accurate and scalable, and consistent performance on held-out countries suggests that it could be used to generate wealth estimates in countries where data are unavailable. Results suggest that such estimates could be used to help target social programs in data poor environments, as well as to understand the determinants of variation in well-being across the developing world.

However, while our CNN-based approach outperforms approaches to poverty prediction that use simple hand-crafted features (e.g. scalar nightlights), the information the CNN is using to make a prediction is less interpretable than these simpler approaches, perhaps inhibiting adoption by the policy community. A key avenue for future research is in improving the interpretability of deep learning models in this context, and in developing approaches to navigate this apparent performance-interpretability tradeoff.

Our deep learning approach is also perhaps best viewed as a way to amplify rather than replace ground-based survey efforts, as local training data can often further improve model performance (Fig. 3b), and because other key livelihood outcomes often measured in surveys -- such as how wealth is distributed

within households, or between households within villages -- are more difficult to observe in imagery. Similarly, our approach could also be applied to the measurement of other key outcomes, including consumption-based poverty metrics or other key livelihood indicators such as health outcomes. Performance in these related domains will depend both on the availability and quality of training data, which remains limited for key outcomes such as consumption in most geographies. Finally, our approach could likely be further improved by the incorporation of higher-resolution optical and radar imagery now becoming available at near daily frequency (Fig. 1b), or in combination with data from other passive sensors such as mobile phones (Blumenstock et al 2015) or social media platforms (Sheehan et al 2019). All represent scalable opportunities to expand the accuracy and timeliness of data on key economic indicators in the developing world, and could accelerate progress towards measuring and achieving global development goals."

• Are there other experiments that would strengthen the paper further? How much would they improve it, and how difficult are they likely to be?

There was a lot of additional statistical analyses done to examine accuracy. However, I would prefer to see either some additional analysis or discussion around the following two points:

1. Why the asset index weights were derived from the entire sample and no consideration was given to deriving weights from individual countries or different time periods. The asset indices could look very different if they were constructed in different ways and therefore this could have significant impacts on the predictive outcomes. If a policy maker was to look at this work currently it would appear that they could achieve 70% accuracy. However, if they wanted to repeat the analysis for a specific country the accuracy they would get would be different.

Response: This is a reasonable concern. In new Fig. S2, we now construct the asset index in different ways, including comparing our pooled PCA to (a) a measure that is just the sum of all the assets owned, (b) a PCA constructed from only objects that are owned (e.g. TV, radio) and not from housing quality scores which are more subjective, and (c) country-specific asset indices created from running the PCA on each country separately. As shown in Fig. S2, correlations between the pooled PCA index we use and these alternative variants range from $r=0.89$ to $r=0.99$.

We now note these results in the main text:

"Alternative methods of constructing the index yield very similar wealth estimates (Fig. S2)".

In Methods we have added the text:

"The PCA-based index is quite robust to methods of calculation as well as variables included in the index. We compare our cross-country pooled PCA index to (a) a measure that is just the sum of all the assets owned, (b) a PCA constructed from only objects that are owned (e.g. TV, radio) and not from housing quality scores which are more subjective, and (c) country-specific asset indices created from running the PCA on each country separately. As shown in Fig. S2, correlations between the pooled PCA index we use and these alternative variants range from $r=0.89$ to $r=0.99$."

2. Its currently not clear to a wide enough audience how the change in assets over time are related to changes in EO data metrics. If this could be focused on it would be were novel as previous works

by other researchers have not looked at the relationships between changes in poverty/well-being and EO data

Response: To better understand what the over-time model is picking up in imagery, we have now visualized model derived features (as we did for the cross sectional comparisons). As in the cross sectional comparisons, the over-time model also appears to pick up changes to agricultural regions and changes in urbanization. This is shown in new Fig S10, where we show the pair of before+after input images and the corresponding activation map. We have also added additional text to the methods section clarifying how the over-time model is trained.

We emphasize that attaching semantic meaning to the features that the model learns is subjective - e.g. the model has not been told to look for urban areas or changes in agricultural activity, but uses changes that (to us) appear related to urbanization or agriculture to make predictions. The goal of the model is to maximize predictive performance on held-out data, not to maximize interpretability of the predictions it makes, and this performance/interpretability tradeoff is an important one in our approach. We now more explicitly highlight this tradeoff in the discussion:

“However, while our CNN-based approach outperforms approaches to poverty prediction that use simple hand-crafted features (e.g. scalar nightlights), the information the CNN is using to make a prediction is less interpretable than these simpler approaches, perhaps inhibiting adoption by the policy community. A key avenue for future research is in improving the interpretability of deep learning models in this context, and in developing approaches to navigate this apparent performance-interpretability tradeoff.”

• Are the claims appropriately discussed in the context of previous literature?

My major question about the claims is that Why only use Median reflectance over 3 years from Landsat? What theoretical basis is there for reflectance being linked to poverty. In other studies land use or land cover is used which is related to poverty by for example saying that building roof material is an indicator of well-being. Policy makers like this approach as it can be related to literature and understanding and other policies they are working on at the time. With reflectance and median reflectance there is less clarity on how it is linked. I know that data science is not as concerned by this but I think some discussion about this would certainly broaden the appeal of the paper.

Response: The reason for using median reflectance over three years is twofold: (1) to reduce the influence of noise and clouds in the imagery, and (2) because we are trying to predict a livelihood value which (unlike consumption) tends to change relatively slowly over time. We have added additional detail in this regard in Methods:

“The motivation for using three-year composites was two-fold. First, multi-year median compositing has seen success in similar applications as a method to gather clear satellite imagery (Azzari et al 2017), and even in 1-year compositing we continued to note the substantial influence of clouds in some regions, given imperfections in the cloud mask. Second, the outcome we are trying to predict (wealth) tends to evolve slowly over time, and we similarly wanted our inputs to not be distorted by seasonal or short-run variation.”

Regarding the theoretical basis for reflectance being linked to poverty, we note that *any* measure that links imagery to poverty is using information in reflectance from the imagery -- including approaches that use hand-crafted features relating to land use or to infrastructure (e.g. satellite-derived building counts or vegetation indices). As noted by the referee, these past approaches have indeed shown that reflectance values (which are sometimes used to generate hand-crafted features) have been shown to be predictive of poverty, and we cite multiple instances of these past efforts in our manuscript.

A key difference between an end-to-end CNN-based approach and a hand-crafted in approach is indeed in the interpretability of the inputs -- i.e. in knowing exactly what reflectance information is being used in the image to make a prediction. We agree with the referee that CNN-based approaches are less immediately interpretable, and model interpretability is an active area of research in the deep learning community. In Fig S5 and new Fig S10, we show that the feature representations learned by the CNNs sometimes do have interpretable meanings. As we state in the main text:

“Visualization of model-derived features suggests that the model learns semantically-meaningful features that are intuitively related to wealth, including filters for urban areas, agricultural regions, and water bodies (Fig. S5).”

We have now also added text to the discussion to highlight this performance-vs-interpretability tradeoff, which we agree is important and a key direction for future research:

“However, while our CNN-based approach outperforms approaches to poverty prediction that use simple hand-crafted features (e.g. scalar nightlights), the information the CNN is using to make a prediction is less interpretable than these simpler approaches, perhaps inhibiting adoption by the policy community. A key avenue for future research is in improving the interpretability of deep learning models in this context, and in developing approaches to navigate this apparent performance-interpretability tradeoff.”

The other previous literature is covered mostly as far as I am aware. But the change over time could be emphasised a little more.

Response: We agree and have now noted the changes over time results in the abstract, and we have moved up the discussion of the over-time results in the main text.

• If the manuscript is unacceptable in its present form, does the study seem sufficiently promising that the authors should be encouraged to consider a resubmission in the future?

Yes, I would say that a resubmission is an excellent approach here as it is certainly worth considering once the clarity of the message is improved.

Response: Thanks again for the thoughtful comments and we hope our revisions address your concerns.

Reviewer #2 (Remarks to the Author):

Review of the manuscript entitled “Using publicly available satellite imagery and deep learning to measure and understand economic well-being in Africa.” This manuscript seeks to use publicly available remote sensing data to map economic well being in Africa. The major claims of this paper are that it provides a state of the art methodology for mapping poverty over space and time in multiple African countries. Their results, while similar to their previous results in Jean et al. 2016, appear to be slightly better and use data sets that are more conducive to global research. Additionally, they have added a change in wealth over time component by performing a panel analysis and have expanded to more countries. These results are novel and are of interest to a wide array of researchers within the field. This paper would influence thinking within the field of big data and poverty as this is a first at this spatial scale and over time. There are some major and minor comments that do arise and are given below. The major issue with this paper is that they are using a vague representation of both wealth and the values they are extracting from remotely sensed images. These issues are described in questions 1, 2, and 3 below. This does not allow a policy maker to really understand what is driving poverty and actually poverty levels. Finally, I don’t feel confident in evaluating the appropriateness and validity of the statistical analysis within the data and suggest that an expert in CNNs evaluate this manuscript.

Response: We are glad the referee found our work novel and of interest to a wide audience, and appreciate the suggestions for improvement, which we respond to below.

Major questions:

1. How does their asset index compare to consumption estimates of poverty that the World Bank generally uses? Can they compare their estimates to at least one country so that it can be known what the differences are in these two methods of poverty estimations and how their results would compare?

Response: This is a good suggestion. Our main reasons for using the wealth index were twofold: a long literature suggests that asset ownership is an as good or better measure of longer-run well-being than consumption (consumption is highly temporally variable), and the fact that we happen to observe village-level asset wealth in harmonized surveys across dozens of African countries; such data are only available for consumption in a handful of countries.

In new analysis, we have now compared our consumption and wealth estimates in separate LSMS data where, in a few countries, data on both assets and on aggregated consumption expenditure were available. These comparisons are shown in new Fig. S3, and we find strong correlations between available consumption and asset data in the three countries where both are readily available ($r^2=0.50$, $r=0.71$). We now note this result in the main text and in the methods. E.g. the methods text now reads:

“While our asset data cannot be used to directly construct poverty estimates -- standard poverty measures are constructed from consumption expenditure data, which are not available in DHS surveys -- household consumption aggregates are available in a subset of the LSMS data just described. Across six surveys in three countries, we find our constructed wealth index is fairly strongly correlated with log surveyed

consumption at the village level, with a weighted r^2 of 0.50 (Fig. S3). This suggests that our approach to wealth prediction could perhaps be useful for consumption prediction as well, particularly as additional consumption data become available to train deep learning models.”

We also now note that our measurement of asset wealth is robust to various approaches of constructing the wealth index (Fig. S2 and Methods), which we believe provides additional evidence that our measure is picking up meaningful differences in wealth across households.

Additionally, we now note in the discussion/conclusion the importance of extending our approach to related development outcomes, including the direct measurement of poverty, and how these could contribute to the measurement and achievement of global development goals:

“Our approach is also perhaps best viewed as a way to amplify rather than replace ground-based survey efforts, as local training data can often further improve model performance (Fig. 3b), and because other key livelihood outcomes often measured in surveys -- such as how wealth is distributed within households, or between households within villages -- are more difficult to observe in imagery. Similarly, our approach could also be applied to the measurement of other key outcomes, including the direct measurement of poverty or other key livelihood indicators. Performance in these related domains will depend both on the availability and quality of training data, which remains limited for key outcomes such as consumption in most geographies. Finally, our approach could likely be further improved by the incorporation of higher-resolution optical and radar imagery now becoming available at near daily frequency (Fig. 1b), or in combination with data from other passive sensors such as mobile phones or social media platforms. All represent scalable opportunities to expand the accuracy and timeliness of data on key economic indicators in the developing world, and could accelerate progress towards measuring and understanding how to achieve global development goals.”

Finally, we re-emphasize that asset indices on their own are very commonly used for both welfare analysis and “poverty targeting” (see Filmer and Scott 2012 paper that we cite). We have added additional text to the main text on this point:

“We focus on asset wealth rather than other welfare measurements (e.g. consumption expenditure) as asset wealth is thought to be a less-noisy measure of households' longer-run economic well-being (Sahn and Stiefel 2003, Filmer and Scott 2012), is a common component of “multi-dimensional” poverty measures used by development practitioners around the world and is actively used as a means to target social programs (Alkire et al 2015, Filmer and Scott 2012), and is much more widely observed in publicly-available georeferenced African survey data.”

2.Sustainable Development Goal #1.1 is to eradicate extreme poverty everywhere. This is currently measured as people living on less than \$1.25 a day. Can the results from this study be used to determine where these people are and if they are being lifted out of extreme poverty?

Response: As described above, we do not directly measure consumption expenditure-based poverty, and thus cannot directly estimate whether villages (or households) are above or below a specific consumption-based poverty line. We see poverty estimation as a key avenue for future work, but believe it to be

additionally challenging for a number of reasons, including because consumption data are often measured with large amounts of noise, and they are unavailable in most countries at the local level.

As noted above, we have added a quantitative comparison between asset wealth and consumption estimates in 3 countries where both are available, and added additional text describing the utility of wealth measures on their own (they are often used directly for targeting and for welfare measurement) and on the importance of future work looking directly at consumption expenditure measures.

3. What values extracted from the satellite data can be used to understand what the model is using to predict variability in poverty so that policy makers can make informed decisions?

Response: This comment relates closely to a comment by R1 on interpretability. While the information that the CNN is using to make predictions is not directly interpretable, some of the features that the model learns are important for wealth prediction can be visualized and interpreted. In Fig S5 (and new Fig S10 on the over-time model), we show that the feature representations learned by the CNNs sometimes do have interpretable meanings. As we state in the main text:

“Visualization of model-derived features suggests that the model learns semantically-meaningful features that are intuitively related to wealth, including filters for urban areas, agricultural regions, and water bodies (Fig. S5).”

However, there is certainly a limit to this interpretability, and an ongoing challenge in this approach is how to maximize predictive performance on held-out data (which our CNN based approach does) while optimally trading off interpretability. We have now also added text to the discussion to highlight this performance-vs-interpretability tradeoff, which we agree is important and a key direction for future research:

“However, while our CNN-based approach outperforms approaches to poverty prediction that use simple hand-crafted features (e.g. scalar nightlights), the information the CNN is using to make a prediction is less interpretable than these simpler approaches, perhaps inhibiting adoption by the policy community. A key avenue for future research is in improving the interpretability of deep learning models in this context, and in developing approaches to navigate this apparent performance-interpretability tradeoff.”

4. Why is a 6.72 km pixel used in this analysis? Also, if this is the pixel sized used, why does the map of Nigeria in figure 4, have 7.5 km pixels?

Response: The Landsat satellite images we use have 30m/pixel resolution. Images were exported as 255x255 px ($30\text{m}/\text{px} * 255\text{px} = 7.65\text{km}$), and then cropped to 224x224 ($30\text{m}/\text{px} * 224\text{px} = 6.72\text{km}$) as 224x224 is the input size for our CNN architecture. A prediction is then made for the entire 255x255 image, and therefore our generated raster for Nigeria in Fig 4 is 7.65km resolution.

These image size choices were due to a technical limitation at the time of original image generation in Google Earth Engine; it was a convenient and somewhat arbitrary choice at the time that allowed us to

explore different deep learning architectures. We have now explained these choices more clearly in Methods.

5. Why are three years of Landsat data used to cover an area? That appears to be long time in rapidly changing countries such as these.

Response: The reason for using median reflectance over three years is twofold: (1) to reduce the influence of noise and clouds in the imagery, and (2) because we are trying to predict a livelihood value which (unlike consumption) tends to change relatively slowly over time. We have added additional detail in this regard in Methods:

“The motivation for using three-year composites was two-fold. First, multi-year median compositing has seen success in similar applications as a method to gather clear satellite imagery (Azzari et al 2017), and even in 1-year compositing we continued to note the substantial influence of clouds in some regions, given imperfections in the cloud mask. Second, the outcome we are trying to predict (wealth) tends to evolve slowly over time, and we similarly wanted our inputs to not be distorted by seasonal or short-run variation.”

6. Why were just DHS data compared out of country and not LSMS? Also, how comparable are the DHS and LSMS data sets? It appears that multiple countries have both. Can one be predicted with the other?

Response: The reason why we did not train models for predicting LSMS out-of-country is because we had much less LSMS data than for DHS. Specifically, we only had ~1,400 LSMS clusters split across 5 countries, compared to nearly 20,000 DHS clusters split across 23 countries. With fewer countries, the model is much more likely to overfit to the countries in the training set, so we decided to let our LSMS models “see” some data in the test country, taking care however that the models are not trained on any satellite image containing the test clusters.

Unfortunately DHS and LSMS data have different sampling frames, and the villages that appear in DHS are not the same as those that appear in LSMS. Thus, the datasets cannot be directly compared—although of course we have been careful to create the LSMS asset index exactly as we have done in DHS. However, what we can do is aggregate both up to the district level and compare values at that level, and doing so gives a correlation of $r=0.77$ ($r^2=0.6$), which we regard as quite high given that the sample is not stratified at the district level (i.e. it is not a random sample of villages within each district; typically it is a random sample at the state or national level).

We have added this comparison of DHS and LSMS indices to the Methods section, indicating that the indices in countries which have data from both LSMS and DHS (aggregated to the district level) have a fairly high r^2 of 0.6:

“LSMS data is processed to try to match our DHS index by matching asset quality definitions as similarly as possible. ... While we cannot directly compare DHS and LSMS indices at the village level, district level estimates from the two sources have an r^2 of 0.6.”

7. In the results, Kenya appears to have the poorest performance. Can this be explained?

Response: We have generated new Fig S11 that explores the national and subnational predictors of performance across countries. Our strongest predictor of performance is the within-village variance in wealth, and our model (which is trained to predict differences between rather than within villages, since we only have village-level geocoordinates) performs somewhat worse in settings where within-village variation is high. Kenya has some of the highest within-village wealth variation in our dataset, and we hypothesize that this is why our Kenyan numbers are somewhat worse than other countries. These results are now discussed in the main text in the added section on “Understanding model performance”.

8. When the Night-time lights and Landsat CNN models are used to create their “combined” model using ridge regression (p. 10 lines 331-336), how many independent variables are there going into the regression? Why is Ridge regression used versus Elastic Net that incorporates LASSO as well?

Response: The “combined” model concatenates the final layer outputs of the separate multispectral (MS) and nightlights (NL) convolutional neural networks (CNNs). The final layer of the CNNs are a 512-dimensional vector, so the combined vector is 1024-dimensional. The final layers of the individual MS-only and NL-only neural networks are subject to L_2 -regularization (along with all other layers in the neural network). Ridge regression (which is linear regression with L_2 -regularization) is used in the combined model such that the combined model mimics a CNN that is joined at the final layer. While Elastic Net would be a valid choice as well, L_2 -regularization (also known as weight-decay) is more commonly used in deep learning settings.

9. The only statistical values that are given for model performance are r-squared. What are the model error estimates and how do they vary spatially and between countries?

Response: In Fig. S4 we now also report RMSE by country.

Minor comments:

1. In Figure s1, use a different range of colors so wealthy and poor can be more easily identified.

Response: Done, thanks for the suggestion.

2. In Figure 4, where are the Local Government Area boundaries and population data from used to create this figure?

Response: Boundaries are sourced from the Database of Global Administrative Areas (GADM), which is now referenced in the caption of Fig. 4.

Reviewers' comments:

Reviewer #1 (Remarks to the Author):

I will keep this report brief as it is just focused on the additions that have been made since the previous draft.

Thank you very much for addressing my comments from the previous draft of the manuscript that I reviewed and apologies for my slow review. The paper is an excellent addition to the study area and is as far as I am aware the first to begin looking at changes overtime in development. I very much appreciate your additional comments and explanation about some of the weaknesses of the current work and how they could be looked at in the future. These weaknesses though should not prevent the manuscript from being published.

There are a couple of comments I have added to the new manuscript. Including suggestions but these are minor. All of these are posted as comments in the pdf. A couple of points here:

page 5 line 166: in locations with heterogeneous wealth within villages the Ground Surveys are going to perform worse because the GPS points for the clusters are displaced. So if villages are heterogeneous the landscape characteristics are likely to be different. So moving these points up to 10km away from their actual locations will mean that the remote sensing data you are using may not be as representative of the landscape characteristics surrounding the communities. I think you could mention this and it would strengthen your results here especially for the lay reader that may not be aware of these issues,

the use of a 7km neighbourhood around the cluster is a new addition to the paper and I am not sure why 7km was chosen. I would explain this as a lot of DHS users will be interested to know the reasoning behind this.

some of the technical language still crosses into jargon for the wider audience, I can see in some places you have attempted to explain in simple terms the data science and ML language. but if you were to add in for everything it would probably make the manuscript too long!

other than that I would just like to thank you for an excellent study! and I look forward to trying out some of these methods using the code you have provided.

Reviewer #2 (Remarks to the Author):

I have read through the comments and their additions to manuscript. I am fine with most of the comments and am supportive of the additions. There are a few minor comments below. I do have one major comment after looking at figure S4. The RMSE values seem awfully high. If the range in values for your asset index is -1 to 1 and you have a mean RMSE of 0.7 or above in some countries are those values awfully high. Would this not indicate that your model is not all that useful predictions? Does it not mean the models have been overfit to get artificially high r-squared values? It appears that you are using 1,024 variables to predict one, so overfitting could be happening.

Minor comments:

Page 7: Line 240: Not sure what a "hand-crafted" feature is. Yes, people use simple features, but not sure they craft them by hand. Please revise.

Figure 2: What is the weighted and un-weighted on e and f? Please add to the figure caption.

Figure S5: What about D?

Reviewer #1 (Remarks to the Author):

Thank you very much for addressing my comments from the previous draft of the manuscript that i reviewed and apologies for my slow review. The paper is an excellent addition to the study area and is as far as i am aware the first to begin looking at changes overtime in development. I very much appreciate your additional comments and explanation about some of the weaknesses of the current work and how they could be looked at in the future. These weaknesses though should not prevent the manuscript from being published.

Response: We again appreciate the reviewer's kind words!

There are a couple of comments i have added to the new manuscript. Including suggestions but these are minor. All of these are posted as comments in the pdf. A couple of points here:

page 5 line 166: in locations with heterogeneous wealth within villages the Ground Surveys are going to perform worse because the gps points for the clusters are displaced. So if villages are heterogeneous the landscape characteristics are likely to be

different. SO moving these points upto 10km away from their actual locations will mean that the remote sensing data you are using may not be as representative of the landscape characteristics surrounding the communities. I think you could mention this and it would strengthen your results here especially for the lay reader that may not be aware of these issues,

Response: Good point - we have not added that the added noise might be particularly consequential in heterogeneous environments (manuscript line 162). Thanks!

The use of a 7km neighbourhood around the cluster is a new addition to the paper and i am not sure why 7km was chosen. i would explain this as a lot of DHS users will be interested to know the reasoning behind this.

Response: The 7km number comes from rounding. The paper has been clarified to more accurately state "6.72x6.72km neighborhood". The 6.72km x 6.72km is a result of cropping images to be the correct input size for our CNN architecture, which takes 224 pixel x 224 pixel inputs. With 30m/px landsat inputs, we have $224 \times 30\text{m} = 6.72\text{km}$.

We have now added a line to explain this more clearly in the main text (paragraph starting line 175) and in the methods.

Some of the technical language still crosses into jargon for the wider audience, i can see in some places you have attempted to explain in simple terms the data science and ML language. but if you were to add in for everything it would probably make the manuscript too long! Other than that i would just like to thank you for an excellent study! and i look forward to trying out some of these methods using the code you have provided.

Response: We again appreciate the reviewer's kind words!

P3 line 63 "We pool all households in our sample in the principle components estimation such that the derived index is consistent over both space and time, and then average household values to the "cluster" level"

You are assuming that the same assets characterise poverty in each country?

That is correct and we have now made that assumption explicit on line 62, and also highlighted that alternate measures that allow country- and year-specific mappings from assets to the wealth index are highly correlated with our pooled wealth index (as shown in Fig S3).

**P4 line 128 “We first use repeated rounds of DHS surveys and spatially match a cluster in one survey year to the nearest cluster in a previous survey year, and compute wealth changes as the difference in wealth index between matched pairs of clusters.”
*did you consider the GPS displacement of clusters when you did this?***

Response: No, we could see no obvious way of accounting for the unknown displacement. We have made this clear on line 127 in the revised manuscript.

Reviewer #2 (Remarks to the Author):

I do have one major comment after looking at figure S4. The RMSE values seem awfully high. If the range in values for your asset index is -1 to 1 and you have a mean RMSE of 0.7 or above in some countries are those values awfully high. Would this not indicate that your model is not all that useful predictions? Does it not mean the models have been overfit to get artificially high r-squared values? It appears that you are using 1,024 variables to predict one, so overfitting could be happening.

Our fault for not making this more clear. Our wealth data in the survey are roughly mean=0, sd=1 at the household level (reducing to sd=0.81 when we aggregate to the village level). The average RMSE at the village level between satellite-based predictions and survey measures is 0.46, so about half a standard deviation, and at the district level is roughly a quarter a standard deviation on average. This is about the same RMSE we get when we compare two different ground-based survey measures of the same outcome (DHS vs census), as noted in the caption. We have now clarified the range of the underlying data in the caption for Fig S4 to make this more clear.

Regarding overfitting, 1024 variables is indeed a large number given the size of our dataset. This is why we use ridge regression, i.e. linear regression with L_2 -regularization, as mentioned in the Methods section (see lines 397 and 450). In particular, the regularization constant is determined using cross-validation to minimize overfitting.

Our best evidence that we are in fact not overfitting comes from evaluating our model performance on held-out survey data, as shown in Figs 2 and S4, and by comparing how satellite predictions do predicting our survey data relative to how independent ground data (from censuses) would do predicting the same survey data. The fact that satellites do just about as well as census data in predicting held-out DHS survey data (i.e. data the satellite model was not trained on) suggests that performance is good and not a result of overfitting.

Minor comments:

Page 7: Line 240: Not sure what a “hand-crafted” feature is. Yes, people use simple features, but not sure they craft them by hand. Please revise.

We have revised to use “simpler” rather than hand-crafted. Thanks.

Figure 2: What is the weighted and un-weighted on e and f? Please add to the figure caption.

Weights are the number of clusters in each district. We have updated the caption in Figure 1 to clarify this, thanks.

Figure S5: What about D?

Good catch - D consists of features that appear to activate in desert terrain. We have made this update to the figure caption.

REVIEWERS' COMMENTS:

Reviewer #1 (Remarks to the Author):

Thank you for addressing the comments and questions i had in the previous draft. I also thank you for address the comments of reviewer 2 and the response letter is nice and clear. I have no more comments to make on this now as i think it is a nice paper that significantly adds to the scientific discipline.

Reviewer #2 (Remarks to the Author):

I am fine with their comments and have no further issues with the manuscript.

Reviewer #1 (Remarks to the Author):

Thank you for addressing the comments and questions i had in the previous draft. I also thank you for address the comments of reviewer 2 and the response letter is nice and clear. I have no more comments to make on this now as i think it is a nice paper that significantly adds to the scientific discipline.

Reviewer #2 (Remarks to the Author):

I am fine with their comments and have no further issues with the manuscript.

We thank both reviewers very much for their thoughtful comments and advice throughout!